# Optimal and efficient text counterfactuals using Graph Neural Networks

**Dimitris Lymperopoulos, Maria Lymperaiou, Giorgos Filandrianos, Giorgos Stamou**
Artificial Intelligence and Learning Systems Laboratory
School of Electrical and Computer Engineering
National Technical University of Athens
jimlibo13@gmail.com, {marialymp, geofila}@islab.ntua.gr, gstam@cs.ntua.gr

## Abstract

As NLP models become increasingly integral to decision-making processes, the need for explainability and interpretability has become paramount. In this work, we propose a framework that achieves the aforementioned by generating semantically edited inputs, known as *counterfactual interventions*, which change the model prediction, thus providing a form of counterfactual explanations for the model. We frame the search for optimal counterfactual interventions as a graph assignment problem and employ a GNN to solve it, thus achieving high efficiency. We test our framework on two NLP tasks - binary sentiment classification and topic classification - and show that the generated edits are contrastive, fluent and minimal, while the whole process remains significantly faster than other state-of-the-art counterfactual editors. [1]

## 1 Introduction

Since the introduction of the Transformer (Vaswani et al., 2017) the field of NLP has enjoyed an abundance of impressive implementations targeting a variety of linguistic tasks. Explainability (Alammar, 2021; Danilevsky et al., 2020) and interpretability (Madsen et al., 2022) in NLP are topics of increasing popularity, researching biases and spurious correlations which hinder the generalization capabilities of state-of-the-art (SoTA) models. Adversarial attacks (Zhang et al., 2020) can trigger alternative outcomes of NLP models unveiling inner workings, therefore providing post-hoc interpetability. Several prior attempts in creating adversarially perturbed inputs, focused on label-flipping scenarios, have been presented in recent literature (Michel et al., 2019; Morris et al., 2020; Li et al., 2020; Ross et al., 2021), while other general-purpose approaches (Ross et al., 2022; Wu et al., 2021) attempt to generate more generic perturbations.

These methods though are accompanied with shortcomings, despite producing promising results in linguistic terms. One practical constraint is that they are computationally expensive (Ross et al., 2021) and relatively slow during inference (i.e. MiCE requires more than 47 hours to produce edits for 1000 samples[2]). Another emerging issue is the fact that diverging from generalized textual generation towards interpretability requires a far more controlled generation process, as the opaque behavior of general-purpose editors (Wu et al., 2021; Ross et al., 2022) built upon Large Language Models (LLMs) often leads to sub-optimal substitutions (Filandrianos et al., 2023) (or at least we have no evidence regarding their optimality and *why* they were selected). In fact, creating optimal linguistic interventions is an algorithmically challenging problem, requiring efficient optimization of the search space of alternatives (Zang et al., 2020; Wang et al., 2021; Lymperaiou et al., 2022; Yin and Neubig, 2022).

In this work, we focus on word-level counterfactual interventions to test the behaviour of textual classifiers when different words are perturbed. Our proposal revolves around placing all implemented interventions under a framework which presents the following characteristics regarding interventions:

- Optimality: Substitutions should be optimal -or approximately optimal-, respecting a given notion of semantic distance.

- Controllability: at least one input semantic should be substituted in each data sample.

- Efficiency: an optimal solution should be reached using non-exhaustive search techniques among alternative substitutions.

We approach these requirements by viewing counterfactual interventions as a combinatorial optimization problem, solvable via graph assignment

---

[1] Code available at https://github.com/Jimlibo/GNN-Counterfactual-Editor

[2] This is concluded through our experimentation.

algorithms from graph theory (Yan et al., 2016). To further enhance our method, we consider the use of Graph Neural Networks (GNNs) (Wu et al., 2019) as a faster approximate substitute of these algorithms (Yow and Luo, 2022). Our proposed method can be applied to both model-specific and general purpose scenarios, since there is no strict reliance on changing the final label. This property allows for generated edits to be used for different tasks apart from label-flipping, such as semantic similarity (Lymperaiou et al., 2022) or untargeted generation (Wu et al., 2021); nevertheless, in this paper, we focus on classification tasks for direct comparison with prior work. To this end, we compare our approach with two SoTA editors (Wu et al., 2021; Ross et al., 2021) using appropriate metrics for label-flipping, fluency and semantic closeness. Approaches based on Large Language Models (Chen et al., 2023; Sachdeva et al., 2024) are not considered in this work, due to their hardware requirements [3]. To sum up our contributions are:

- We impose optimality and controllability of word interventions translating them in finding the optimal assignment between graph nodes.

- We accelerate the assignment process by training GNNs on these deterministic matchings, ultimately achieving advanced efficiency.

- Our highly efficient black-box counterfactual editor consistently delivers SoTA performance compared to existing white-box and black-box methods on two diverse datasets and across four distinct metrics. Remarkably, it achieves these results in less than 2% and 20% of the time required by its two competitors, demonstrating both superior efficacy and efficiency.

- The versatility of our proposed editor is demonstrated in different scenarios, since it is able to be optimized towards a specific metric or perform general-purpose fluent edits.

## 2 Related work

Exposing vulnerabilities present in SoTA models has been an active area of research (Szegedy et al., 2014), endorsing the probing of opaque models through adversarial/counterfactual inputs.

Granularity of perturbations ranges from character (Ebrahimi et al., 2018) to word level (Garg and Ramakrishnan, 2020; Ren et al., 2019) or even sentence level (Jia and Liang, 2017). In our work, we focus on semantic changes, following the paradigm of word-level perturbations.

Manual creation of adversarial examples has been explored (Gardner et al., 2020; Kaushik et al., 2020; Mozes et al., 2022) with the purpose of changing the true label. Automatic text generation initially implemented via paraphrases (Iyyer et al., 2018), and most recently using masked language modelling (Li et al., 2021; Ross et al., 2021; Li et al., 2020), targets predicted label changes in binary/multi-label classification or textual entailment setups. Similarity-driven substitutions based on word embedding distance (Jin et al., 2020; Zhu et al., 2023) ensure optimality in local level for classification tasks, while constraint perturbations guarantee controllability of adversarials (Morris et al., 2020). Those works partially preserve some desiderata of our approach; however, they are model-specific and thus constrained. General purpose counterfactual generators fine-tune LLMs to offer diverse perturbations, applicable in multiple granularities (Wu et al., 2021; Gilo and Markovitch, 2022; Ross et al., 2022). Prompting on LLMs opens novel trajectories for textual counterfactuals (Chen et al., 2023; Sachdeva et al., 2024), even though explainability of interventions is completely sacrificed, due to the unpredictability of LLM decision-making. Overall, utilizing LLMs is computationally expensive, while produced substitutions may not be optimal as far as word distance is concerned (Filandrianos et al., 2023). On the other hand, interventions through the use of graph-related optimizations (Zang et al., 2020; Lymperaiou et al., 2022) have recently emerged, showcasing that advanced performance and explainability of interventions are on par with computational efficiency.

## 3 Problem formulation

The basis of our work constitutes a graph-based structure that places words extracted from sentences on nodes, and their in-between substitution costs on edges. Let's consider a bipartite graph $G = (V, E)$, where the edge set $E$ consists of all the weighted edges in the graph, and the node set $V$ consists of the source set $S$ of cardinality $|S| = n$ and the target set $T$ of cardinality $|T| = m$, such that $S \cup T = V$, $S \cap T = \emptyset$. Finding optimal

---

[3] Quantization of the LLMs used in these works could alleviate the problem at the cost of performance. Experimentation with this claim is left for future work

connections between nodes of $G$ has been a long sought discrete optimization problem of graph theory, where the optimal match for each node $s \in S$ needs to be determined among a predefined candidate set of nodes $t \in T$. Assuming that $W$ denotes the edge weight set consisting of the weights of all edges $e \in E$, a *min weight matching* $M \subseteq E$ searches for a subset of the lightest possible sum of edge weights $\sum w_e, w_e > 0 \in W$ containing those edges $e \in E$ that cover all nodes of the $min(|S|, |T|)$ set of $G$. Therefore, in the case of $|S| \leq |T|$, all nodes in $S$ will be substituted[4], should an outcoming edge $e_{s \to t}$ exists from each $s$ to any $t \neq s$, denoting that this substitution is feasible. Under these requirements, we formulate the following constraint optimization problem:

$$min \sum w_e, \; subject \; to \; s \neq t \; if \; \exists e_{s \to t} \quad (1)$$

A naive solution to this constraint optimization problem would be the exhaustive search of all possible $(s, t)$ combinations, by examining all possible $m!$ permutations of $T$ until the optimal solution of $min \sum w_e$ is reached. This yields an exponential complexity of $O(m^n)$ (proof in App. D), supposing that $G$ is complete, i.e. each pair of $s - t$ nodes is connected so that $E = S \times T, |E| = nm$. Nevertheless, computational efficiency compared to the naive approach is guaranteed if we view this constraint optimization problem as a variant of the rectangular linear assignment problem (RLAP) (Bijsterbosch and Volgenant, 2010): $n$ source nodes should be assigned to $m \geq n$ target nodes optimally, so that the total weight of the assignment is minimized. RLAP also allows multiple matchings to each source node $s$ thus providing more flexibility in optimal matchings. Assignment algorithms borrowed from older literature (Kuhn, 1955; Karp, 1978) are adapted to solve RLAP, achieving best deterministic complexity of $O(mn \log n)$, significantly reducing the exponential $O(m^n)$.

### 3.1 Graph neural network for RLAP

Graph Neural Networks (GNNs) (Scarselli et al., 2009) have emerged as a powerful tool for learning representations of graph-structured data, making them particularly well-suited for applications in which relationships between entities can be naturally expressed as graphs. In the context of linear assignment problems (Burkard and Çela, 1999),

---

4 These guarantees are explained in Section 4.2

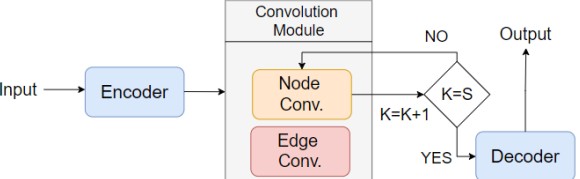

Figure 1: The architecture of the proposed GNN model. In the node convolution layer, node attributes are updated for a total of $S \geq 2$ iterations.

a GNN is employed to solve the linear sum assignment problem (LSAP), where $n$ agents need to be assigned to $n$ jobs under one-to-one matching constraints, while the cumulative cost remains minimal (Liu et al., 2024). Inspired by this approach, we adopt and slightly modify the proposed framework by harnessing a Graph Convolutional Network (GCN) (Kipf and Welling, 2017) to accommodate RLAP; to the best of our knowledge, no prior work has leveraged GNN modules to solve RLAP. The described GCN model consists of three modules: the encoder, the convolution module and the decoder (Figure 1).

#### 3.1.1 Encoder/Decoder

Given the bipartite graph $G$, the encoder module applies a Multi-Layer Perceptron (MLP) to each edge to transform the attributes of the constructed graph into latent representations, thus forming the embedding features. Note that initially the attribute of each edge is simply its weight so that $e_{ij} = w_{ij}$, where $e_{ij}$ denotes the attributes of the edge connecting nodes $i$ and $j$ and $w_{ij}$ is the weight of this edge. Also, the raw attributes of the nodes are initialized as zero-valued vectors. The transformed graph is then passed to the convolutional module as input to update its state. The decoder coupled with the encoder reads out the edge attributes from the output graph and predicts each edge label through an update function. Similarly, the update function is designed as an MLP and mapped to each edge to form edge labels through a sigmoid activation.

#### 3.1.2 The convolution module

The convolution module is comprised of a node convolution layer and an edge convolution layer. For the $i^{th}$ node in the graph, the node convolution layer collects the information from adjacent edges and its $1^{st}$ order neighboring nodes by adaptive aggregation weights and updates its attributes. For each edge, the edge convolution layer aggregates the attribute vectors of the two nodes that

the edge connects, and updates the edge attribute vector. Although the reception field of the convolution module regards $1^{st}$-order neighborhoods, the messages on each node can reach all other nodes after two iterations of convolution, since the graph is bipartite consisting of two node sets (see Section 3), and each node from one set connects with all other nodes of the other set. As a result, the reception field of the convolution module can cover the whole graph after the $2^{nd}$ iteration.

The edge convolution layer first collects information about each edge based on its two adjacent nodes using the aggregation function:

$$\overline{e}_{ij} = [v_i \odot c^u, v_j \odot c^u, e_{ij} \odot c^e] \qquad (2)$$

where $e_{ij}$ denotes the attributes of the edge connecting node $i$ and node $j$, $v_i$ and $v_j$ the attributes of $i^{th}$ and $i^{th}$ nodes and $\odot$ indicates the element-wise multiplication of two vectors. The operator $[\cdot, \cdot, \cdot]$ concatenates its input vectors channel-wise, while the vectors $c^u$ and $c^e$ are the node and edge channel attention vectors with the same dimensions as node attributes and edge attributes respectively. We must also clarify that $\overline{e}_{ij}$ is an intermediate vector representing the concatenated features the edge $i \to j$ and not the *updated edge attribute vector*. After the aggregation function, an update function $\rho^e$ designed as an MLP takes the concatenated features as input and outputs the updated feature, so that: $e_{ij} \leftarrow \rho^e(\overline{e}_{ij})$.

The node convolution layer collects information from adjacent edges and $1^{st}$-order neighborhoods for each node. Specifically, for the $i^{th}$ node in the bipartite graph $G$ we apply the following function:

$$\overline{v}_i = \frac{1}{N_i} \sum_{j=1}^{N_i} \rho_1^v([e_{ij} \odot c^e, w_{ij}(v_j \odot c^u)]), \quad (3)$$

$$e_{ij} \in \mathcal{E}_i \text{ and } v_j \in \mathcal{V}_i$$

where $\rho_1^v$ denotes the function to transform its input to an embedding feature. $\mathcal{E}_i$ denotes the attribute set of all edges associated with node $v_i$ in $G$, and $\mathcal{V}_i$ represents the attribute set of $1^{st}$-order adjacent nodes to node $v_i$. For node $v_i$, $w_{ij}$ is the weight measuring the contribution of its adjacent node $v_j$ during feature aggregation, and is computed as $w_{ij} = \tau([v_i, v_j])$. The collected embedding features are then concatenated with the current attributes of node $v_i$ and are passed to another transformation function that outputs the updated attributes for node $v_i$ using the formula

$v_i \leftarrow \rho_2^u([\overline{v}_i, v_i])$. Functions $\rho_1^v$, $\rho_2^v$ and $\tau$ are all specified as MLP modules, each of them with a different architecture and parameters[5].

# 4 Counterfactual generation overview

The workflow of our method (Figure 2) comprises of three stages. A textual dataset $D$ serves as the input to our workflow. In the first stage, words are extracted from $D$, based on their part of speech (POS), and used as the source node set $S$. The target set $T$ is either a copy of $S$, or else produced from an external lexical source such as WordNet (Miller, 1995), containing all possible candidate substitutions of the source words (nodes). The $S$ and $T$ sets form a bipartite graph $G$ (described in Section 3), with their in-between edge weights reflecting word similarity. In the second stage, we pass the constructed $G$ as input to the trained GCN which outputs an approximate RLAP solution, in the form of a list of candidate word pairs. Each word pair, consists of the source word $s_i \in S$ and its computed substitution $t_i \in T$. In the third and final stage, we harness beam search to define the final changes. Beam search uses a *heuristic function* to choose the most suitable substitutions from those returned by the GNN. The selected words from $S$ are then substituted with their respective pair from $T$, producing a counterfactual dataset $D^*$.

## 4.1 Graph creation

When constructing the bipartite $G$, words are extracted from the original $D$ based on their POS. To test how well our framework generalizes, we use both POS-specific and POS-agnostic word extraction. The former means that we only select to potentially change words that belong to a specific POS (i.e. adjectives, nouns, verbs, etc.), while the latter means that we regard all words, irrespective of their POS. For the edge weights, we employ two different approaches, each varying in transparency. For the first one, we adopt a fully transparent approach by calculating the distances using a lexical hierarchy: the weight of an edge connecting two words is determined by their similarity value as defined in WordNet.[6] In the second case, we apply different LLMs to generate word embeddings, namely AnglE[7] (Li and Li, 2023; Sean et al., 2024), GISTEm-

---

[5]For more information refer to Liu et al. (2024), where they explain in-depth the model architecture and parameters.

[6]path_similarity function between synsets corresponding to the words (https://www.nltk.org/howto/wordnet.html).

[7]mixedbread-ai/mxbai-embed-large-v1

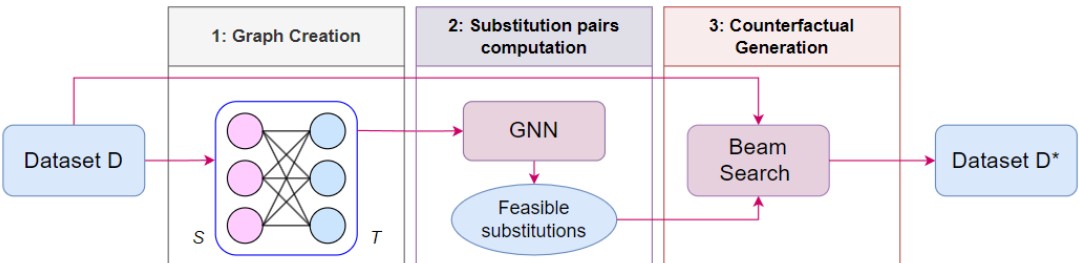

Figure 2: The pipeline of our method. In the first stage, we construct a bipartite graph using words as nodes, and in the second stage we utilize a GNN to get feasible substitutions that approximately solve the RLAP. In the final stage, we use beam search to change appropriate words of the original dataset, thus getting a new counterfactual dataset.

bed[8] (Solatorio, 2024), JinaAI[9] (Mohr et al., 2024) and MUG[10]; then, we set the edge weight equal to the cosine similarity of the two word embedding vectors. Since lower similarity is associated with lighter edges, i.e. more suitable candidates for $M$, the selected words to be substituted will form *contrastive word pairs*. In order to preserve syntax and human readability in the POS-agnostic case, we force substitutions between same-POS words exclusively: thus, we experiment with an edge filtering mechanism, which sets a predefined large weight to edges, $\sim$10 times bigger than the normal edge weights as instructed from WordNet path similarity or cosine similarity of embeddings. This way, we avoid cases where a POS is substituted with a word of different POS, since a significantly heavier edge cannot be selected to participate in $M$. In the POS-specific case, this mechanism is redundant since all words are of the same POS.

### 4.2 Substitution pairs computation

For appropriate substitution pairs we need to solve RLAP on the constructed graph $G$. As previously discussed (Section 3), traditional deterministic approaches achieve this in $O(mn \log n)$. While these methods provide the optimal solution, they lack speed as the dataset size, and therefore graph size grows larger. In an attempt to produce substitution pairs in *stable* time regardless dataset size, we use a GNN model (Section 3.1), which approximates the optimal solution found by deterministic algorithms, while significantly speeding up the process. This way, **efficiency** is guaranteed. By solving the problem with the constraint of minimum $\sum w_e$, we find all most *dissimilar* $s \to t$ pairs, achieving *approximate* **optimality** of concept substitution within $G$

and ultimately producing contrastive substitution pairs. At the same time, **controllability** is *partially* ensured since the graph $G$ is dense (therefore there are no disconnected $s$ nodes) and $|S| \leq |T|$, since $T$ is either a copy of $S$ or produced based on $S$ using antonyms from WordNet (more than one antonym may correspond to each word). Note here, that we use the word *"partially"* as there is a trade-off between *controllability* and *minimality* [11] (see App. A), which stems from using beam search during counterfactual generation. In practice, there are also a few exceptions in controllability, if a source concept cannot be mapped on WordNet.

### 4.3 Counterfactual Generation

As a result of solving RLAP, a matching $M \subset E$ is returned, indicating the optimal substitutions to $n$ source concepts. We denote as $W_n^M \subset W$ the total weight of $M$ that contains $n$ source concepts. Given this matching, beam search selects which conceptual substitutions from $M$ will *actually* be performed on $D$. This selection process is necessary since we desire changes to be *minimal* in terms of number of words altered per instance, perturbing only small portions of input, a property which has been argued to make explanations more intelligible (Alvarez-Melis et al., 2019; Miller, 2019). In this context, we also set an upper limit of substitutions on each text instance, experimenting with both a fixed and a dynamically set number. In the second case, for each instance, the upper limit is equal to the $20\%$ of the total number of words it contains. We stop the search when the model's prediction is flipped or when the upper limit is reached, thus keeping the number of edits low.

[8]avsolatorio/GIST-Embedding-v0
[9]https://jina.ai/embeddings/
[10]Labib11/MUG-B-1.6

[11]Minimality here refers to the number of words changed.

# 5 Experiments

In this section, we present our experiments along with the results, which showcase that our framework produces fluent, minimal edits with high label-flipping percentage in a short amount of time compared to the other editors. All experiments were run on the same system consisting of a *16 GB GPU*, an *Intel i7 CPU* and *16 GB RAM*.

## 5.1 Experimental Setup

**Datasets** We evaluate our framework and compare it with other editors from literature, on two English-language datasets: IMDB, which contains movie reviews and is used for binary sentiment classification (Maas et al., 2011) and a 6-class version of the 20 Newsgroups used for topic classification (Lang, 1995). Due to the high computational demands of the compared methods, we sampled 1K instances from each dataset for evaluation. Running MiCE on just 1K samples required over 47 hours (see Table 1), making full dataset experiments impractical. We chose twice the sample size used in similar studies comparing the same methods on the same datasets (Filandrianos et al., 2023).

**Predictors** We test our edits using the same predictor models with MiCE (Ross et al., 2021) in each dataset. These models are based on RoBERTa$_{LARGE}$ (Liu et al., 2019) and boast a test accuracy of 95.9% and 85.3% for IMDB and Newsgroups respectively.

**Editors** We compare our framework with two SoTA editors, MiCE (Ross et al., 2021) and Polyjuice (Wu et al., 2021). MiCE produces minimal edits optimized for label-flipping, while Polyjuice is a general purpose editor, whose edits are not restricted to a specific task. In regard to our framework, we use the approach of the deterministic RLAP solution as a baseline, and we compare it with the GNN RLAP optimization. To test the generalization properties of our work, we also use POS-restricted and POS-unrestricted substitutions.

**Metrics** To assess the performance of the different editors, we draw inspiration from MiCE and measure the following properties: (1) **flip-rate:** the percentage of instances for which an edit results in different model prediction (label-flipping); (2) **minimality:** the "size" of the edit as measured by word-level Levenshtein distance between the original and edited input. We adopt a normalized version of this metric with a range of [0, 1] — the Levenshtein distance divided by the number of words in the

original input; (3) **closeness:** the semantic similarity between the original and edited input, measured by BERTscore (Zhang et al., 2019); (4) **fluency:** a measure of how similarly distributed the edited input is compared to the original. To evaluate fluency, we first take a pretrained T5-BASE model (Raffel et al., 2020) and compute the loss value for both the edited and original input. Afterwards, we report their *loss_ratio* - i.e., *edited / original*. Since we aim for a value of 1.0, which indicates equivalent losses for the original and edited texts, the final measure of fluency is defined as $|1 - loss\_ratio|$.

## 5.2 Results

The results of our experiments are shown in Table 1, including both IMDB and Newsgroups datasets. More analysis can be found in App. A, B.

Our proposed editors—deterministic and GNN-powered—outperform both MiCE and Polyjuice across the three of the four metrics namely **minimality**, **fluency** and **closeness**. Regarding flip-rate, MiCE achieves the highest results (99% - 100%, across the two datasets), followed by our approach: our best editor reaches values slightly above 90% (specifically 94.4% for IMDB and 92% for Newsgroups). However, this is expected, since MiCE is the only editor that has white-box access to the classifier and it is able to strategically construct edits that affect the classifier the most, regardless of the input text.

Results also show that our edits tend to be more minimal when graph construction is based on embeddings models instead of WordNet (approximately 10% of the original tokens are changed when WordNet is employed, while with embedding models only 1% of the said tokens change). We believe this is due to the fact that SoTA embedding models are able to better depict concept distance compared to WordNet, and therefore substitutions based on them are of higher quality, leading to more contrastive pairs. This means that for the same impact on the classifier's output, less embedding substitutions are required compared to WordNet-based ones. On the other hand, using embedding models reduces the overall transparency of the method. Despite minor discrepancies, all our framework variants consistently outperform previous techniques across every metric for Polyjuice and three metrics for MiCE. Moreover, even the general-purpose variation of our framework, which lacks access to the classifier, yields better results compared to the white-box MiCE, in just 2% of the time.

| IMDB | | | | | |
|---|---|---|---|---|---|
| **Editor** | **Fluency ↓** | **Closeness ↑** | **Flip Rate ↑** | **Minimality ↓** | **Runtime ↓** |
| Deterministic w. fluency | 0.14 | 0.969 | 0.892 | 0.08 | 4:09:41 |
| GNN w. fluency | 0.07 | 0.986 | 0.861 | 0.12 | 3:17:51 |
| GNN w. fluency & dynamic thresh | 0.057 | 0.986 | 0.851 | 0.146 | 4:18:34 |
| GNN w. fluency & POS_filter | 0.08 | 0.992 | 0.862 | 0.123 | **0:32:05** |
| GNN w. fluency & edge filter | 0.105 | 0.993 | 0.845 | 0.149 | 3:00:38 |
| GNN w. fluency_contrastive | 0.112 | **0.999** | 0.914 | 0.014 | 2:12:06 |
| GNN w. contrastive | 0.048 | 0.996 | 0.927 | **0.01** | 2:00:15 |
| GNN w. AnglE & contrastive | 0.063 | 0.995 | 0.944 | 0.011 | 0:45:38 |
| GNN w. GIST & contrastive | 0.037 | 0.995 | 0.882 | 0.016 | 0:58:14 |
| GNN w. JinaAI & contrastive | 0.047 | 0.995 | 0.928 | 0.017 | 1:00:56 |
| GNN w. MUG & contrastive | **0.036** | 0.996 | 0.889 | 0.013 | 0:52:19 |
| Polyjuice | 0.394 | 0.787 | 0.782 | 0.705 | 5:01:58 |
| MiCE | 0.201 | 0.949 | **1.000** | 0.173 | 48:37:56 |

| Newsgroups | | | | | |
|---|---|---|---|---|---|
| **Editor** | **Fluency ↓** | **Closeness ↑** | **Flip Rate ↑** | **Minimality ↓** | **Runtime ↓** |
| Deterministic w. fluency | 0.182 | 0.951 | 0.870 | 0.135 | 4:20:52 |
| GNN w. fluency | 0.074 | 0.985 | 0.826 | 0.151 | 3:48:37 |
| GNN w. fluency & dynamic thresh | 0.043 | 0.984 | 0.823 | 0.148 | 4:47:14 |
| GNN w. fluency & POS filter | 0.044 | 0.989 | 0.841 | 0.143 | 1:19:57 |
| GNN w. fluency & edge filter | 0.12 | 0.989 | 0.834 | 0.151 | 3:05:08 |
| GNN w. fluency_contrastive | 0.088 | 0.979 | 0.875 | 0.033 | 2:45:31 |
| GNN w. contrastive | 0.033 | 0.989 | 0.920 | 0.033 | 2:02:34 |
| GNN w. AnglE & contrastive | 0.005 | 0.995 | 0.904 | 0.027 | 1:09:13 |
| GNN w. GIST & contrastive | **0.001** | 0.995 | 0.898 | 0.02 | 1:02:55 |
| GNN w. JinaAI & contrastive | 0.013 | 0.993 | 0.882 | 0.025 | 0:57:31 |
| GNN w. MUG & contrastive | 0.005 | **0.996** | 0.900 | **0.016** | **0:53:04** |
| Polyjuice | 1.153 | 0.667 | 0.8 | 0.997 | 6:00:10 |
| MiCE | 0.152 | 0.922 | **0.992** | 0.261 | 47:23:35 |

Table 1: Experimental results of counterfactual generation. We evaluate different versions of our framework using the metrics described on subsection 5.1, and we compare it with MiCE and Polyjuice. For each metric (column) the best value is highlighted in **bold**. Reported runtimes refer to inference.

As far as runtime is concerned, our editors show a remarkable improvement in speed compared to MiCE and Polyjuice. Our deterministic editor, which is used as a baseline, requires approximately 4 hours for each dataset, while editors that use the GNN discussed in Section 3.1 achieve faster execution on average (2-4 hours). Runtime is further improved with the use of embedding models, where execution requires less than an hour (52 minutes - 1 hour for IMDB, 53 minutes - 1 hour and 9 minutes for Newsgroups). This significant speed improvement is one of the main advantages of our framework compared to the two SoTA editors, where we observed approximately 97% and 83% speed improvement compared with MiCE and Polyjuice respectively.

**Static vs. Dynamic Threshold**   To keep the number of edits relatively low, a way to limit the number of substitutions per data instance is required, accepting a potential drop in flip-rate. For this reason, we use two different approaches. In the first one, we enforce a static number of maximum substitu-

tions allowed for each textual input, regardless of its length; after experimentation, the best number was found to be 10. In the second approach, we dynamically compute the optimal upper limit (or threshold) of substitutions based on the total number of words in the text. After different attempts, we end up defining that limit as 20% of the total number of words. Results however, show insignificant improvement in metrics when using dynamic threshold, while the runtime is increased (approximately by 1 hour per dataset). This slow-down is expected since dynamic threshold introduces an extra linear complexity for each text instance, in place of the O(1) complexity of the static case. Static is our default approach unless stated otherwise.

**POS-restricted vs. Unrestricted Substitutions**
In an attempt to evaluate our editor's ability to distinguish which POS is more influential to a specific dataset when related words are substituted, we impose restrictions regarding which POS should be candidates for substitutions, and compare the results with a POS-unrestricted version of our frame-

work. The IMDB dataset is used for sentiment classification, and therefore adjectives and adverbs are presumed to mainly dictate the label (sentiment) for each instance (Benamara et al., 2005). With that in mind, we limit our editor to change only those two POS. Newsgroups is a dataset which belongs to the topic classification category. Since a topic is deduced by examining the nouns in a text, we instruct the editor to take into account only those. As we observe from Table 1, both editors, with and without POS filtering, achieve very similar results. This holds true for both IMDB and Newsgroups datasets, showing that the observed similarity is not due to a specific POS restriction. The only significant difference is seen in runtime (32 - 60 minutes for restricted editors, 2 - 4 hours for unrestricted ones), which is to be expected since when we only consider certain POS at a time, we also limit the amount of words that will be considered as candidates for substitution. This means that the graph nodes and edges of $G$ will be significantly reduced, thus decreasing the time needed for graph construction and GNN inference.

**Edge Filtering** In order to preserve the POS in each substitution, we apply a penalty mechanism (filtering) when computing edge weights of the graph. This mechanism assigns a weight approximately $10\times$ bigger than the *normal* weights (as defined from WordNet path similarity or embedding cosine similarity), to each edge that connects different-POS words. This way, since our framework is trying to find a *minimum weight matching*, edges with large weights are almost impossible to be chosen and therefore substitutions involving different POS have a low occurrence probability. By examining the results with and without the use of edge filtering we observe that they are quite similar. This leads us to assume that such a mechanism is redundant and its functionality is covered by the GNN solution to our graph assignment problem.

**Contrastive vs fluent contrastive edits** Since the selection of eligible substitutions is a general-purpose process (only defined by the graph), we examine the behaviour of our editor when optimized for label-flipping scenarios. This optimization is done by altering the heuristic function of beam search in the last stage of our framework (see Figure 2). For general-purpose edits, this function is the metric for *fluency* discussed in Subsection 5.1, which assists the production of fluent edits. For label flipping, we use *contrastive probability*,

which regards the change to the model prediction for the original label, to determine the best edits (see *GNN w. contrastive* in Table 1). Finally, we also use the average of fluency and contrastive probability as the heuristic function, which results in fluent edits with high flip-rate (see *GNN w. fluency_contrastive* in Table 1). While the general-purpose edits achieve the lowest flip-rate, they remain better in all metrics compared to Polyjuice, another general-purpose editor. This shows that our framework can also be used as a general, untargeted editor with high-quality edits (regarding discussed metrics); extensive experimentation on this claim is left for future work. The label-flipping optimized edits, achieve better results in *fluency, closeness and minimality* compared to MiCE, a SoTA white-box editor optimized for label-flipping. Therefore, in terms of flip-rate, MiCE demonstrates superior performance, exceeding ours by 7%, accepting a significant 20x slowdown in execution.

**WordNet vs. Embeddings** We investigate the effect of using cosine similarity of embeddings in place of WordNet path similarity between two words, when computing the weight of a specific edge in the bipartite graph $G$. On the one hand, determinstc hierarchies provide more *explainable* relationships between concepts, fully justifying causal pathways of substitutions. On the other hand, recently-emerged embedding models can better capture the relationship and similarity of two words, compared to WordNet. To keep our framework relatively lightweight, we deploy the top four best performing models that participated in an embedding benchmark competition (Muennighoff et al., 2023) and whose size does not exceed *1.25 GB*. Models with that size occupied the top spots in the competition and any increase in model size did not result in significant improvements in performance. Results justify our assumptions, with our variants that leverage the embedding models achieving better results in all metrics compared to our WordNet-based variants. Regarding *GPU inference*, the embedding models also outperform WordNet in terms of speed, since the latter requires API calls for each word/graph node of $V$, which greatly slow down the graph creation process.

## 6 Conclusion

In this work, we present a framework for generating optimal and controllable word-level counterfactuals via graph-based substitutions, which we

evaluate on two classification tasks. We introduce a GNN approach that enhances our proposed baseline deterministic graph assignment algorithm and significantly speeds up the process overall. We compare our results with two SoTA editors, and show that we surpass them in most metrics, while being considerably faster. As future work, we consider integrating more external lexical sources (e.g. ConceptNet) to enhance the possible substitution candidates, as well as improving the performance of the GNN model used to solve RLAP to further approximate deterministic optimal solutions. Other future directions include comparison with LLM-based counterfactual editors and evaluation on other NLP tasks apart from classification.

## Broader Impacts and Ethics

Our framework is intended to aid the interpretation of NLP models. As a model-agnostic explanation method by design (not optimized towards a certain metric in the default case), it has the potential to impact NLP system development across a wide range of models and tasks. In particular, our edits can assist developers working on the NLP field in facilitating, debugging and exposing model vulnerabilities. The framework can also assist in data augmentation which results in less biased and more robust systems. As a consequence, downstream users of NLP models can also be benefited by gaining access to those systems.

While our work focuses on interpreting NLP models, it could be misused in other contexts. For instance, malicious users might generate adversarial examples, such as slightly altered hate speech, to bypass toxic language detectors. Additionally, using these editors for data augmentation could inadvertently lead to less robust and more biased models, as the edits are designed to expose model weaknesses. To avoid reinforcing existing biases, researchers should carefully consider how they select and label edited instances when using them for training. However, such threats are applicable to any text editor in NLP literature and are not tailored on our work.

## Limitations

Our framework comes with its challenges. One of them is that it requires a strong enough GPU (at least 8GB based on our experiments) to run the GNN and the embedding models. Such hardware may not be available to any researcher that wishes to reproduce our experiments. Another one, is the dependence of word existence in WordNet, in cases it serves as a knowledge base for $T$ construction, or as a means for calculating path similarity. For example, if a word from the original input does not exist in the WordNet hierarchy, then we are unable to find its antonyms and therefore a substitution on that word may not occur. The usage of other knowledge sources, which could potentially resolve this limitation, is left for future work. The usage of embeddings for concept distance definition partially resolves the WordNet limitation, even though it results in a slight decrease in explainability of edits: the WordNet structure is well-defined and deterministic, while the model mapping words onto an embedding space does not come with inherent guarantees of its functionality. Explainability is also decreased when using the GNN module in place of the deterministic min weight matching algorithm for solving RLAP (Kuhn, 1955; Karp, 1978), since the reason why an edge (and therefore a candidate substitution pair) is selected becomes less transparent, as a result of a black-box procedure performed by the GNN. Finally, while not being a direct limitation, the general-purpose applicability of our framework has not been presented experimentally in the current paper, despite being a natural consequence stemming from the optimization performed on the graph.

## Acknowledgments

The research work was supported by the Hellenic Foundation for Research and Innovation (HFRI) under the 3rd Call for HFRI PhD Fellowships (Fellowship Number 5537).

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

## A  Trade-offs

Since our editor is a highly customizable one, there are many trade-offs which must be considered during counterfactual generation.

**Controllability vs. Minimality**  Controllable interventions involve changing *any* semantic that can be changed in order to observe an outcome; to this end, we could potentially alter as many words as possible in order to reach a goal, e.g. label-flipping. However, in our case, in order to produce minimal edits, we set a maximum number of substitutions per textual input and leverage *beam search* to select the most appropriate changes. As a consequence, the default controllability requirement is partially sacrificed, since it is not guaranteed that all words that can be substituted will be indeed substituted. Nevertheless, our framework still produces edits for each input, meaning that it will change the original text, although not entirely; this is why we impose as controllability to *modify at least one word of the original data sample*. In our experiments (see Table 1) we have accepted this trade-off since our interest lies more heavily with minimality compared to controllability. Despite that, it is possible to fully ensure controllability by arsing the limitations mentioned above (i.e. max number of substitutions and beam search), although such an approach would results in worse performance regarding minimality.

**Optimality vs. Execution Speed**  In our framework, we use both a deterministic (see *Deterministic w. fluency* from Table 1) and a GNN approach (see *GNN w. fluency* from Table 1) to solve RLAP. With the deterministic approach, optimality is ensured, since traditional graph matching algorithms have been proved to find the optimal solution (Kuhn, 1955; Karp, 1978). However, the complexity of those algorithms, which is $O(mn \log n)$, results to slower runtimes as graph size increases (which is analogous to the number of words to be substituted and therefore depends on the dataset size). By replacing the deterministic algorithms with the trained GNN (see Section 3.1), our framework becomes significantly faster at the cost of optimality. This is due to the fact that the solution given by the GNN is an *approximation* of the optimal one.

**Explainability vs. Execution Speed**  In our work, we utilize WordNet as the default way of computing edge weights between nodes, where each edge weight is based on the path that connects a source word $s$ with target word $t$ in WordNet. By mapping each concept to WordNet synsets, a deterministic concept position is assigned to each word, providing a fully transparent concept mapping to a well-crafted lexical structure. The utilization of

word embeddings casts a shadow on word mapping, since we transit to a vector representation of an uninterpretable multi-dimensional space via black-box models. Similarity in the embedding space translates to semantic similarity of physical concepts, acting as our guarantee towards employing embedding models.

In combination with the deterministic solution to RLAP, WordNet mapping guarantees *explainability* of edits, since all paths $s \rightarrow t$ are tractable, and the choice of edges is fully transparent due to the deterministic selection process of graph matching algorithms (Bijsterbosch and Volgenant, 2010). By obtaining the resulting matching $M$ we gain full access to the set of edits to perform $S \rightarrow T$ transition. A sacrifice in explainability is imposed when using the GNN instead of the deterministic graph assignment algorithms: the GNN introduces an uncertainty to the edge selection, since we cannot be entirely sure *why* a specific edge was chosen. Although we have trained the GNN to output the RLAP solution, the model itself still remains a black-box structure that hides the exact criteria which decide whether an edge will be selected or not. Still, in some applications the speedup offered by the GNN outweighs this drop in explainability, while the opposite may hold in cases where trustworthiness is of utmost importance.

Overall, as observed from our experiments (see Table 1), leveraging embedding models to compute edge weights and the GNN to solve RLAP showcases major improvements in *fluency*, *flip-rate* and *minimality*, while also being considerably faster. Someone could argue that this approach is clearly better that the fully deterministic one, since it produces higher quality edits. Despite that, we need to point out that these improvements come at a significant cost on explainability, since, due to the GNN, the edge selection process is no longer transparent and edge weight computation depends on black-box embedding models.

## B Edits Comparison Between Editors

Qualitative comparisons with Polyjuice and MiCE are presented in this Section to demonstrate the capabilities of our framework regarding minimality and flip-rate. For that purpose, we choose an instance of the IMDB dataset which is originally classified as 'positive' and acquire the edited instances from our framework and the two editors mentioned above. Specifically for Polyjuice, since

its goal is to change the prediction from *positive* to *negative*, we use the control code [negation], which guides the editor to generate an edit that is the negation of the original text. The original along with the edited inputs (red words denote changes made by each editor) are shown in Figure 3.

Figure 3: Original input and edited inputs from different editors. The changes that each editor performed are highlighted in red color.

As we can see, MiCE performs the highest number of interventions on the original input, with two of those changes being semantically incorrect (*"conservative, conservative"* and *"both of whom have"*). We also notice that its changes are not entirely word-level, which further deteriorates the editor's performance regarding *minimality*. Polyjuice on the other hand, makes only one change at the end of the text, which however has no semantic meaning; such edits may betray the presence of a counterfactual editor or a neural model in general, coming in contrast with the requirement of "imperceptible edits" that commonly involves counterfactual interventions. Our editor presents the best performance out of the three, changing only one word, while being semantically correct and very close to the original instance.

Numeric results of Figure 3 instances regarding *minimality* and *label-flipping* are reported in Table 2. Since we only have one textual instance, instead of *flip-rate* we use the term *prediction flipped* to denote whether the edited input is able to change the original prediction of the classifier. Note that Polyjuice is unable to flip the prediction, while both

| Edits | Minimality ↓ | Prediction Flipped |
|---|---|---|
| Polyjuice | 0.078 | False |
| MiCE | 0.256 | **True** |
| Ours | **0.011** | **True** |

Table 2: Metric results of the edits presented in Figure 3. For each property (column) the best value is highlighted in **bold**.

MiCE and our framework succeed. Also, our editor is the best as far as minimality is concerned, with Polyjuice being second and MiCE being the worst out of the three.

## C GNN Training

For training the GNN incorporated in our framework, we commence from the trained model described in Liu et al. (2024) and fine-tune it to our specific problem, which is RLAP. The process we follow is almost identical to the one reported by the authors, with a small difference regarding the loss function being used. Initially, a synthetic dataset that consists of M samples[12] is created. Each sample is composed of a cost matrix C in which the elements are generated from a uniform distribution on $(0, 1)$ and the corresponding optimal assignment solution which is obtained by the Hungarian algorithm (Kuhn, 1955). We consider the RLAP as a binary classification task and divide the elements in the ground-truth assignment matrix $Y^{gt}$ [13] into positive labels and negative ones. Since for each node, there is at most one positive edge among its adjacent edges and the rest are negative ones, we use the *Balanced Cross Entropy* as the loss function, to avoid the negative labels dominating the training:

$$L = -\sum_{i=1}^{n}\sum_{j=1}^{m} \big( w \times y_{ij}^{gt} \log(y_{ij}) + (1-w)\times$$

$$(1 - y_{ij}^{gt})\log(1 - y_{ij}) \big)$$

(4)

where $y_{ij}$ is the predicted label for edge $i \rightarrow j$ which connects source node $i$ and target node $j$, $y_{ij}^{gt}$ is the corresponding ground-truth vector element indicating the edge as positive or negative, and $w$ is the weight which balances the loss to avoid the negative labels dominating the training. Parameters

---

[12]Each sample represents a weighted bipartite graph.
[13]$Y^{gt}$ is a matrix where element $y_{ij}^{gt}$ is 1 if the edge connecting nodes $i$ and $j$ **belongs** to the minimum matching, else it is -1.

$n, m$ denote the cardinality of source and target nodes sets, so that $|S| = n, |T| = m$.

As in Liu et al. (2024), training takes 20 epochs in total, where the learning rate is set as $0.003$ initially and declined by $5\%$ after every 5 epochs.

## D Proof of naive graph matching complexity

We will prove the exponential $O(|T|^{|S|})$ complexity of the naive solution to the constraint optimization problem of adversarial $s - t$ matchings. Given the example graph of Figure 4 with $S = \{A, B, C\}$ of cardinality $|S| = 3$ and $T = \{1, 2, 3, 4\}$ of cardinality $|T| = 4$, the following node combinations occur:

Source node A can take $|T| = 4$ values: A-1, A-2, A-3, A-4. Node B can independently of A take $|T| = 4$ values: B-1, B-2, B-3, B-4. Finally, C independently of A and B can also take $|T| = 4$ values: C-1, C-2, C-3, C-4. Therefore, all combinations for the $|S| = 3$ source nodes are $4 \times 4 \times 4 = 4^3 = |T|^{|S|}$

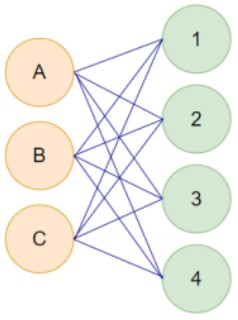

Figure 4: Example graph