# OpenReview forum: "Optimal and efficient text counterfactuals using Graph Neural Networks"
_EMNLP/2024/Workshop/BlackBoxNLP — BlackboxNLP 2024_

### Official Review · Reviewer_jsfc · 2024-09-03

**Overall Assessment:** 4
**Confidence:** 3

**Best Paper:**

1

**Best Paper Justification:**

None

**Comments Questions Suggestions And Typos:**

The abstract could use a few words about the specific approach. There's currently no information whatsoever about how you perform the counterfactual editing as compared to previous works.

**Paper Summary:**

Propose a fast counterfactual editing technique based on a Graph Convolutional Network that solves the linear sum assignment problem to find minimal edits. The GCN returns a list of candidates consisting of source words and substitutions, the most suitable of which is selected with beam search, resulting in an *approximately optimal* substitution while being more efficient than methods creating optimal substitutions.

In the experiments, the paper compares the technique to MiCE (Ross et al., 2021) and Polyjuice (Wu et al., 2021), and shows that it outperforms them in three out of four evaluation metrics (*fluency*, *closeness*, and *minimality*; MiCE performs best in *flip rate*) and in runtime. The experiments also compare various setups of the method, such as using embeddings versus WordNet, and with and without a POS filter.

**Summary Of Strengths:**

Counterfactual evaluations are a growing technique in explainable NLP and model evaluations. Therefore, finding controllable but efficient techniques is a timely topic.

The paper is well-written, well-motivated and the technical details are easy to follow. The trade-offs between different properties of counterfactual editing techniques (controllability, optimality, explainability, runtime) are highlighted.
The proposed technique outperforms others from the literature both in quality measures and in runtime. This makes it valuable for future works that use counterfactual editing.
The experiments are extensive, and the appendix even provides qualitative comparisons between the different editing techniques.

**Summary Of Weaknesses:**

For an explainability workshop, it would have been insightful to also have a demonstration of the method in use. But the substance of the paper is enough for a publication as-is.

---

### Official Review · Reviewer_u4LD · 2024-09-04

**Overall Assessment:** 3
**Confidence:** 4

**Best Paper:**

1

**Best Paper Justification:**

-

**Comments Questions Suggestions And Typos:**

Row 345: Typo, should be "JinaAI"

Q1: In Section 4.1, if in the POS-agnostic case only substitutions between same-POS words are performed (rows 352-354), is the only difference with the POS-specific case that words with different POS can be selected in the same evaluation? Is there experimental evidence that allowing for different-POS words would degrade the replacement validity?

**Paper Summary:**

This work proposes a new framework for crafting word-level counterfactual interventions for text classifiers. The proposed approach frames the task as a variant of the rectangular linear assignment problem (RLAP), employing a Graph Convolutional Network (GCN) to approximate optimal deterministic assignment algorithms. Practically, the GCN is composed by an encoder contextualizing the latent representations of graph elements from their neighbors, and a decoder predicting edge labels from contextualized attributes, optimized end-to-end. The model is trained over a bipartite graph in two settings: one in which the graph edges are derived from WordNet path synset similarity, and one based on embedding similarity. Finally, beam search is used to update selected dataset words with valid alternatives. The approach is compared to Polyjuice and MiCE, existing counterfactual generation approaches, and results show improvements in output fluency and closeness to the original text, with a major improvement in runtime compared to baseline approaches.

**Summary Of Strengths:**

Formulating valid counterfactuals is an important problem in interpretability research, and the proposed method seems like a valid and more rapid alternative to existing non-LLM-based approaches. The work is presented clearly and, to my best knowledge, it represents the first application of graph neural networks to identify feasible substitution in counterfactual generation.

**Summary Of Weaknesses:**

The main downside of this work is the lack of evaluation on LLM-based counterfactual generators such as those described in DISCO (https://aclanthology.org/2023.acl-long.302) and CATfOOD (https://aclanthology.org/2024.eacl-long.113/). While the authors state that "Overall, utilizing LLMs is computationally expensive, while produced substitutions may not be optimal as far as word distance is concerned" (rows 146-149), it would have been useful to have an LLM baseline using a reasonably sized open-source system for Table 1 comparison. While these systems can undoubtedly be computationally expensive to run, a quantized version of e.g. LLaMA 3.1 8B can comfortably run on the resources employed by the authors for their experiments. The stated concern about the minimality of LLM contrastive examples could then be validated empirically and in light of the runtime of the LLM for the set of given examples.

While the concern that the "explainability of interventions is completely sacrificed, due to the unpredictability of LLM decision-making) (row 145) might be motivating the choice of not including such an approach, it can be argued that neural modules such as GNNs for assignment and word embeddings for similarity can render the core components of the proposed pipeline similarly opaque. Referring to pre-LLM approaches like Polyjuice and MiCE as SOTA does not seem reasonable in light of the significant advances in the instruction-following ability of LLMs since 2021, when these works were published.

---

### Official Review · Reviewer_TATG · 2024-09-13

**Overall Assessment:** 3
**Confidence:** 3

**Best Paper:**

1

**Best Paper Justification:**

NA

**Comments Questions Suggestions And Typos:**

NA

**Paper Summary:**

This paper presents a novel framework for generating optimal and controllable word-level counterfactual explanations for NLP models using GNNs and graph assignment algorithms. The authors frame the problem as a rectangular linear assignment problem  and use a GNN to efficiently approximate solutions. They evaluate their approach on sentiment classification and topic classification tasks, comparing it to SoTA counterfactual editors.

**Summary Of Strengths:**

- The proposed method significantly improves computational efficiency compared to SOTA counterfactual editors, achieving comparable or better results in a fraction of the time.
- The paper provides a good theoretical basis for the work, framing the problem as a RLAP and leveraging graph theory concepts.

**Summary Of Weaknesses:**

- Limited dataset scope - the evaluation is conducted on only two datasets (IMDB and 20 Newsgroups). Testing on a wider range of NLP tasks and datasets would strengthen claims of generalizability.
- While automated metrics are used for evaluation, there is no human evaluation of the generated counterfactuals, which could provide insights into their interpretability and usefulness

---

### Decision · Program_Chairs · 2024-09-20

**Decision:**

Accept

**Comment:**

The reviewers appreciate the novelty of the proposed method for generating counterfactual explanations. As such, the work provides a valuable addition to BlackboxNLP. The authors are encouraged, however, to address the weaknesses pointed out by reviewer u4LD, who states that comparing the proposed method to LLM-based counterfactual generators is a missing key ingredient in the paper.